# Degradative Signaling in ATG7-Deficient Skeletal Muscle Following Cardiotoxin Injury

Fasih Ahmad Rahman, Troy Campbell, Darin Bloemberg, Sarah Chapman and Joe Quadrilatero *

Department of Kinesiology and Health Sciences, University of Waterloo, Waterloo, ON N2L 3G1, Canada; fasih.rahman@uwaterloo.ca (F.A.R.); troy.campbell@uwaterloo.ca (T.C.); bloembed@mcmaster.ca (D.B.); chapman.sarah@queensu.ca (S.C.)
* Correspondence: jquadril@uwaterloo.ca

**Abstract:** Skeletal muscle is a complex tissue comprising multinucleated and post-mitotic cells (i.e., myofibers). Given this, skeletal muscle must maintain a fine balance between growth and degradative signals. A major system regulating the remodeling of skeletal muscle is autophagy, where cellular quality control is mediated by the degradation of damaged cellular components. The accumulation of damaged cellular material can result in elevated apoptotic signaling, which is particularly relevant in skeletal muscle given its post-mitotic nature. Luckily, skeletal muscle possesses the unique ability to regenerate in response to injury. It is unknown whether a relationship between autophagy and apoptotic signaling exists in injured skeletal muscle and how autophagy deficiency influences myofiber apoptosis and regeneration. In the present study, we demonstrate that an initial inducible muscle-specific autophagy deficiency does not alter apoptotic signaling following cardiotoxin injury. This finding is presumably due to the re-establishment of ATG7 levels following injury, which may be attributed to the contribution of a functional *Atg7* gene from satellite cells. Furthermore, the re-expression of ATG7 resulted in virtually identical regenerative potential. Overall, our data demonstrate that catastrophic injury may "reset" muscle gene expression via the incorporation of nuclei from satellite cells.

**Keywords:** muscle regeneration; muscle injury; cardiotoxin; autophagy; apoptosis; cell death; satellite cells





## 1. Introduction

Skeletal muscle is a highly resilient tissue that consists of multinucleated, post-mitotic fibers that facilitate movement through contraction. Muscular contractions in the form of exercise or activities of daily living can cause damage to skeletal muscle [1]. Additionally, traumatic injury can also elicit significant damage to skeletal muscle; however, the severity of damage can vary, resulting in the repair or regeneration of myofibers. Mild damage to skeletal muscle may result in plasma membrane rupture, myofibrillar disruption, and Z-disk streaming [2,3], whereas large-scale damage causing the death of myofibers and surrounding structures will require a well-orchestrated regenerative response to produce new functional myofibers [4,5].

In response to damage, a significant degree of intracellular remodeling occurs within skeletal muscle. One of the major mechanisms responsible for the remodeling of skeletal muscle is macroautophagy (hereby referred to as autophagy). Autophagy is a conserved degenerative process responsible for the breakdown of cellular material. Cellular autophagy is a finetuned process, the alterations of which can result in the deterioration of cellular homeostasis and lead to apoptosis. Emerging evidence suggests that autophagy plays an important role in the remodeling of cells, particularly during the differentiation of cultured myoblasts [6–8]. Our lab and others have demonstrated that the lack of autophagy in cultured myoblasts results in increased susceptibility to apoptosis, whereas a mild elevation of autophagy can protect against apoptosis [6–8].

In skeletal muscle tissue, autophagy has been shown to be necessary in the regeneration process. For example, the loss of ATG16L1 and ULK1 or the inhibition of autophagy via 3-methyladenine (3-MA) results in impaired skeletal muscle regeneration following injury [9–11]. Furthermore, autophagy markers have been shown to be elevated several days following muscle injury [9–11]. One of the key events for autophagy to occur is the lipidation of microtubule-associated protein 1 light chain 3 (MAP1LC3 or LC3) by ATG7 to form a mature autophagosome [12]. The lack of mature autophagosome formation results in the accumulation of cellular debris, thereby leading to skeletal muscle decline [13]. Overall, these studies support the notion that autophagy is required for skeletal muscle regeneration. To date, it is unknown whether a relationship between autophagy and apoptosis exists in injured skeletal muscle and whether autophagy deficiency results in augmented apoptotic signaling in myofibers, ultimately leading to diminished skeletal muscle regeneration. Here, we examine the role of skeletal muscle-specific autophagy deficiency and apoptosis throughout the regeneration time course.

## 2. Results

### 2.1. Atg7 Knockout in Skeletal Muscle Reduces Muscle Mass and Autophagic Markers

Absolute muscle mass of three primarily fast-twitch muscles, including the gastrocnemius (GAS), tibialis anterior (TA), and plantaris (PLANT), was reduced in *Atg7^fl/fl^:HSAMCM* knockout compared to *Atg7^fl/fl^* control mice; however, there was no change in the soleus (SOL) mass, a muscle that contains a large percentage of slow-twitch fibers (Table 1). In contrast, relative muscle mass was only reduced in the GAS and PLANT muscles ($p < 0.05$), but not the TA or SOL (Figure 1A–E).

Immunoblot analyses of GAS at D0 (i.e., uninjured muscle) revealed a reduction of approximately 61% in ATG7 protein content in *Atg7^fl/fl^:HSAMCM* compared to *Atg7^fl/fl^* animals (Figure 1F; $p < 0.05$). The residual ATG7 levels observed in knockout muscle is likely a result of other cell populations within the muscle that contain a functional *Atg7* gene, including endothelial cells (i.e., from blood vessels), satellite cells, and/or immune cells [14]. No significant changes in SQSTM1 or LC3BI, but a trend ($p = 0.10$) toward lower LC3BII, was observed between groups (Figure 1G–I). A significant decrease in the LC3BII:I ratio (Figure 1J; $p < 0.05$), indicative of reduced autophagic flux, was observed in the *Atg7^fl/fl^:HSAMCM* animals compared to their control counterparts.

**Table 1.** Body weight and muscle weight descriptives.

| | *Atg7^fl/fl^* | *Atg7^fl/fl^:HSAMCM* |
|---|---|---|
| Body Weight (g) | 26.9 ± 0.52 | 25.1 ± 0.47 |
| **Absolute Muscle Weight (mg):** | | |
| Gastrocnemius | 123.6 ± 1.23 | 106.48 ± 0.89 * |
| Tibialis Anterior | 43.9 ± 0.75 | 39.8 ± 0.61 * |
| Soleus | 7.6 ± 0.25 | 7.4 ± 0.26 |
| Plantaris | 16.7 ± 0.42 | 14.2 ± 0.32 * |
| **Relative Muscle Weight (mg/g):** | | |
| Gastrocnemius | 4.5 ± 0.15 | 4.2 ± 0.16 * |
| Tibialis Anterior | 1.6 ± 0.11 | 1.5 ± 0.10 |
| Soleus | 0.28 ± 0.04 | 0.29 ± 0.06 |
| Plantaris | 0.62 ± 0.05 | 0.50 ± 0.06 * |

Data shown as mean ± SEM. * Denotes significant difference compared to *Atg7^fl/fl^* animals ($p < 0.05$).

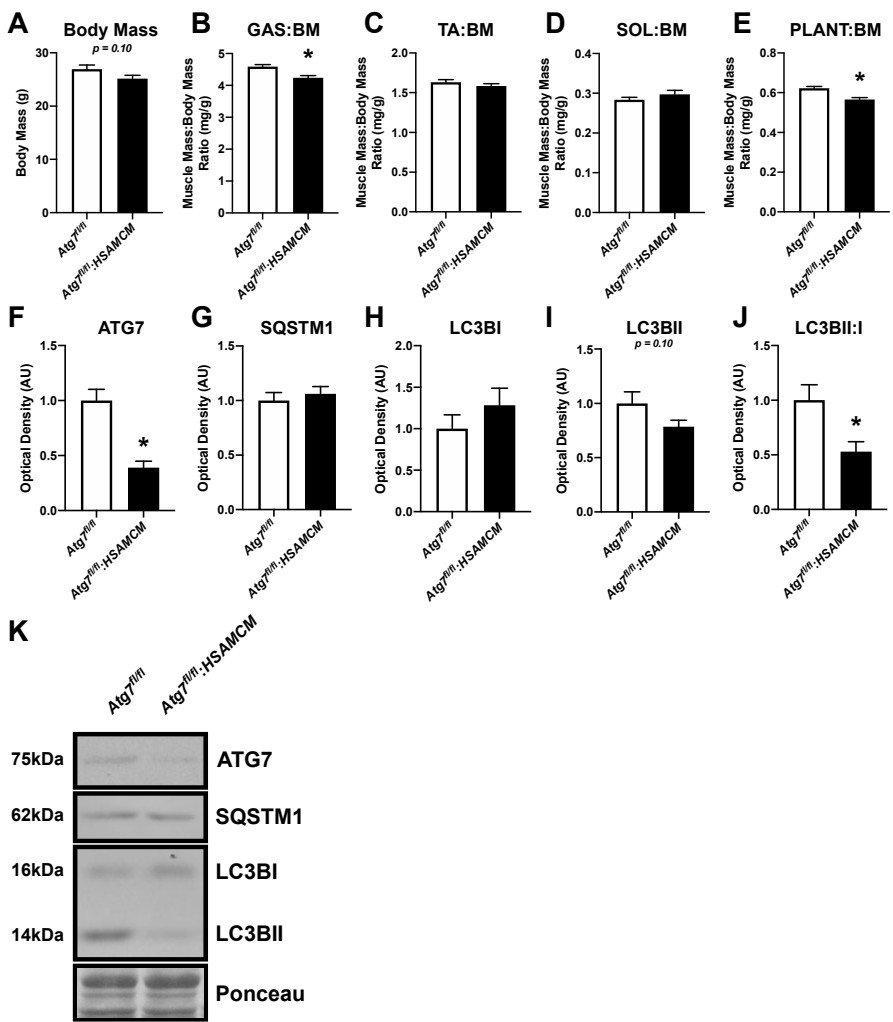

**Figure 1.** Uninjured animal skeletal muscle weights and autophagic protein levels. (**A**) Body mass of *Atg7^{fl/fl}* and *Atg7^{fl/fl}:HSAMCM* animals following 5 days of tamoxifen treatment. (**B–E**) Normalized skeletal muscle mass of the gastrocnemius (GAS), tibialis anterior (TA), soleus (SOL), and plantaris (PLANT). (**F–J**) Quantification of ATG7, SQSTM1, LC3BI, LC3BII, and LC3BII:I ratio. (**K**) Representative immunoblots. * Significant differences compared to *Atg7^{fl/fl}* animals ($p < 0.05$). n = 6–8 per group.

## 2.2. CTX-Injury Does Not Alter Mass Parameters in Skeletal Muscle of Atg7 Knockout Mice

To investigate the role of autophagy following skeletal muscle injury, we induced skeletal muscle damage via intramuscular injection of cardiotoxin (CTX) and collected muscles at day 3 (D3), D7, and D14 following damage. Uninjured muscles are referred to as D0. In response to CTX, control and knockout mice displayed no alterations in body mass (Figure 2A) and a similar recovery of hindlimb muscle mass (Figure 2B–E; $p < 0.05$). Interestingly, the TA:BM ratio demonstrated a main effect of genotype (Figure 2C). To our initial surprise, *Atg7^{fl/fl}:HSAMCM* animals began to express ATG7 in injured muscles (Figure 2F). This was particularly evident at D3 following CTX injury and was several-fold higher compared to uninjured skeletal muscle (Figure 2F,M; main effect of time: $p < 0.05$). Additionally, a rapid decrease in SQSTM1 was observed in both groups at D3, which returned to baseline levels by D14 (Figure 2G,M; main effect of time: $p < 0.05$). Furthermore, LC3BI, LC3BII, and the LC3BII:I ratio generally increased with time (Figure 2H–J,M; main effect of time: $p < 0.05$ for all). To validate whether autophagy was being induced in both *Atg7^{fl/fl}* and *Atg7^{fl/fl}:HSAMCM* animals, the upstream regulator BECN1 and the cysteine protease ATG4B were examined. There was a significant induction of BECN1 and ATG4B following muscle injury (Figure 2K–M; main effect of time: $p < 0.05$ for both). To further investigate ATG7 levels, immunoblotting was performed on undamaged D14 quadriceps muscles.

Indeed, ATG7 levels were significantly lower in uninjured D14 quadricep muscle from *Atg7^{fl/fl}:HSAMCM* animals (Figure 3A,B). In addition, immunofluorescent staining confirmed the lack of ATG7 in undamaged skeletal muscle fibers (D0) and the re-establishment of ATG7 in skeletal muscle fibers at D7 and D14 following damage (Figure 3C). Coupled with the above findings (Figure 2F,M), these findings demonstrate that injury restores ATG7 levels in deficient animals.

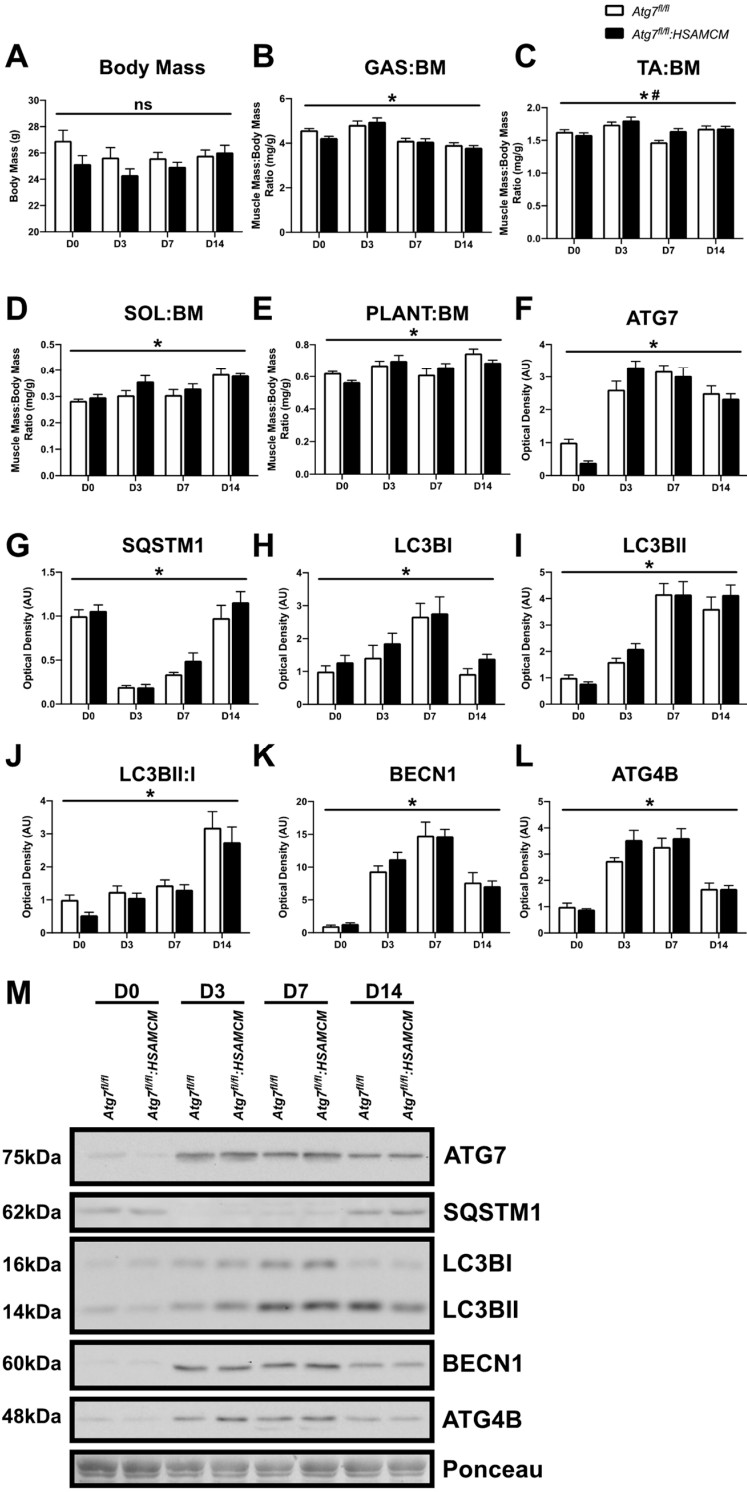

**Figure 2.** Skeletal muscle weights, and autophagic protein levels before (D0) and following (D3, D7, D14) cardiotoxin (CTX)-induced injury. (**A**) Body mass of *Atg7^{fl/fl}* and *Atg7^{fl/fl}:HSAMCM* animals. (**B–E**) Normalized muscle mass measurements of gastrocnemius (GAS), tibialis anterior (TA), soleus

(SOL), and plantaris (PLANT). (**F**–**L**) Quantification of ATG7, SQSTM1, LC3BI, LC3BII, LC3BII:I ratio, BECN1, and ATG4B. (**M**) Representative immunoblots. * Significant main effect of the regenerative time course ($p < 0.05$). # Significant main effect of genotype ($p < 0.05$). ns indicates no significant main effects. n = 7–8 per group.

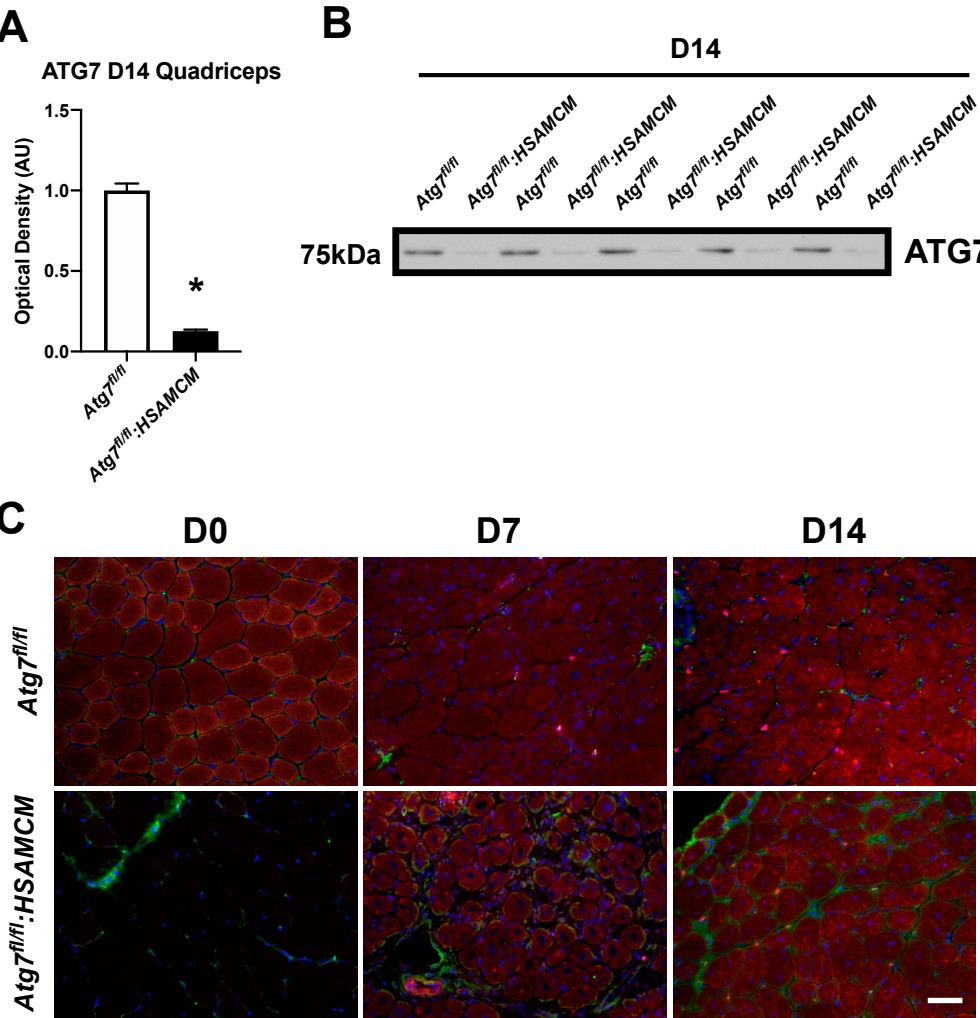

**Figure 3.** ATG7 levels in uninjured and injured quadriceps muscle. (**A**) Quantification and (**B**) representative immunoblot of ATG7 in uninjured quadriceps muscles. CTX was not injected into the quadriceps, only TA and GAS. (**C**) Immunofluorescent staining of TA muscle for PAX7 (green), ATG7 (red), and DAPI (blue). * Denotes significant differences compared to *Atg7<sup>fl/fl</sup>* animals ($p < 0.05$). Scale bar indicates 50 μm. n = 5 per group.

Histological assessment of skeletal muscle regenerative capacity, as measured by calculating the area occupied by regenerating skeletal muscle within the CTX-injured area of each section, revealed an almost identical regenerative pattern in *Atg7<sup>fl/fl</sup>* and *Atg7<sup>fl/fl</sup>:HSAMCM* muscles (Figure 4A,B; main effect of time: $p < 0.05$). D3 muscles in both animals display virtually no regenerating muscle fibers, which increased to approximately 80% by D7 and 100% by D14 (Figure 4B). This is consistent with other studies demonstrating virtually complete regeneration by D7 post-CTX [15].

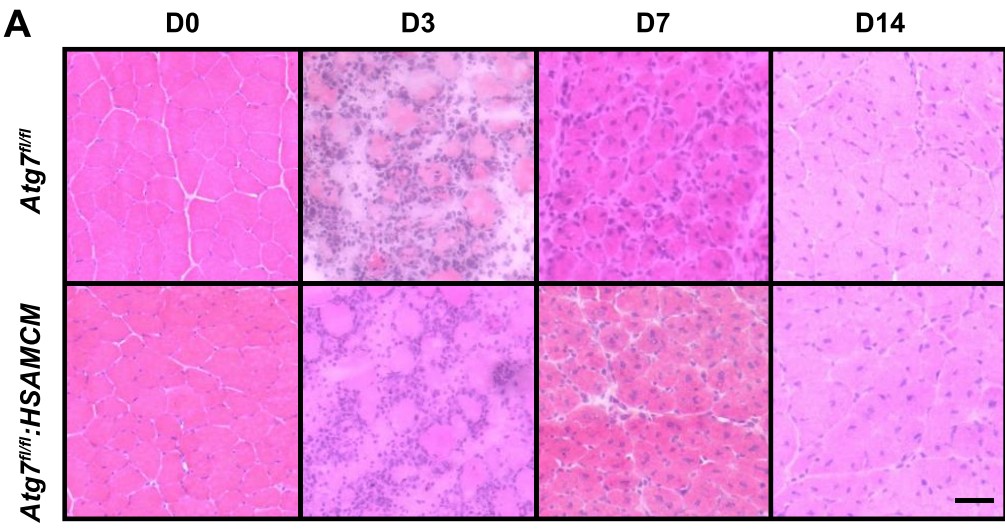

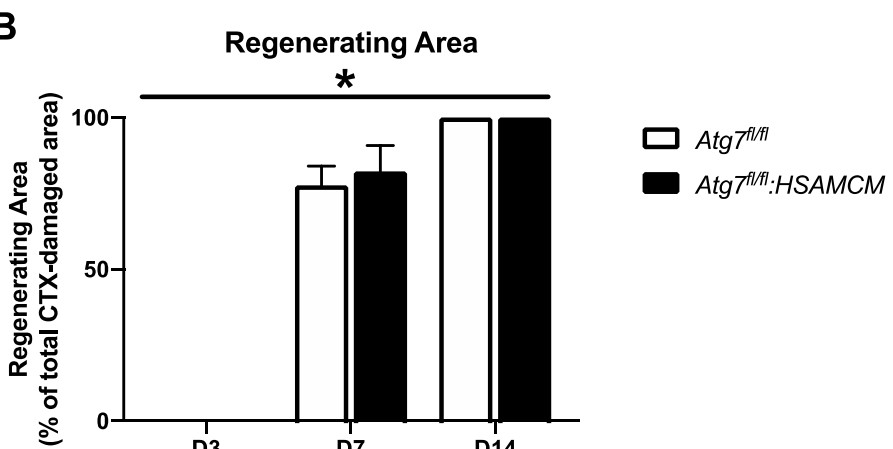

**Figure 4.** Skeletal muscle histology before (D0) and following (D3, D7, D14) CTX-induced injury. (**A**) Representative images of H&E stained cryosections. (**B**) Quantification of regenerating area throughout the regeneration time course. * Significant main effect of the regenerative time course ($p < 0.05$). n = 7–8 per group. Scale bar indicates 50 μm.

*2.3. Fiber-Type Distribution Remains Largely Unchanged but CSA in Glycolytic Fibers Is Slightly Larger in Atg7$^{fl/fl}$:HSAMCM Animals*

Next, we performed a fiber-type distribution analysis on the TA muscle at D0 and D14 to understand basal and end-point differences. Although some significant main effects of time ($p < 0.05$) and genotype ($p < 0.05$) were noted in IIA and IIAX fiber distribution, the effects were minimal. However, there was a larger effect of time ($p < 0.05$) where IIXB distribution increased, while IIB fiber distribution decreased in both groups (Figure 5A,B). Furthermore, the mean cross-sectional area was greater for IIX, IIXB, and IIB fibers in D14 *Atg7$^{fl/fl}$:HSAMCM* compared to D0 and D14 *Atg7$^{fl/fl}$ and Atg7$^{fl/fl}$:HSAMCM* muscles (Figure 5C). We also found a significantly larger mean cross-sectional area of IIB fibers in D0 *Atg7$^{fl/fl}$* muscles compared to D0 *Atg7$^{fl/fl}$:HSAMCM* muscles (Figure 5C).

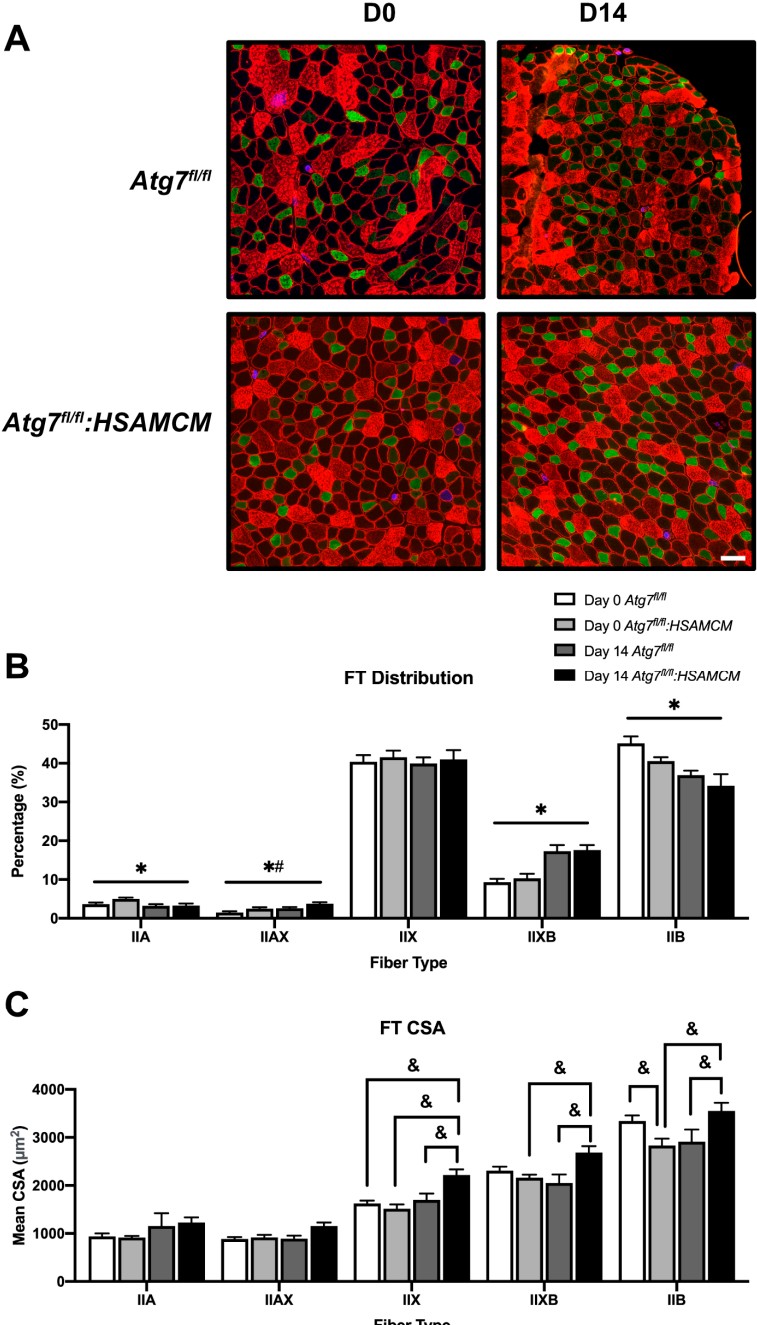

**Figure 5.** Immunofluorescent analyses of fiber-type distribution before (D0) and following (D3, D7, D14). (**A**) Representative images of fluorescently labeled fibers showing type I (blue), type IIA (green), type IIX (unstained), and type IIB (red). Quantification of (**B**) fiber-type distribution and (**C**) fiber-type cross-sectional area. * Significant main effect of the regenerative time course ($p < 0.05$). # Significant main effect of genotype ($p < 0.05$). & Significant interaction effect between individual groups ($p < 0.05$). n = 7–8 per group. Scale bar indicates 50 μm.

## 2.4. Atg7 Knockout Does Not Impact Atrophic and Apoptotic Protein Markers in Response to CTX Injury

Although ATG7 was restored following CTX-induced injury in our knockout mice, we questioned whether there were any atrophic and apoptotic alterations as a consequence of the initial knockout. The levels of two major muscle-specific E3 ubiquitin ligases, TRIM63 and FBXO32, were dramatically reduced in response to injury. Although levels displayed a

significant ($p < 0.05$) time-dependent increase in both groups (Figure 6A,B), they did not reach baseline levels.

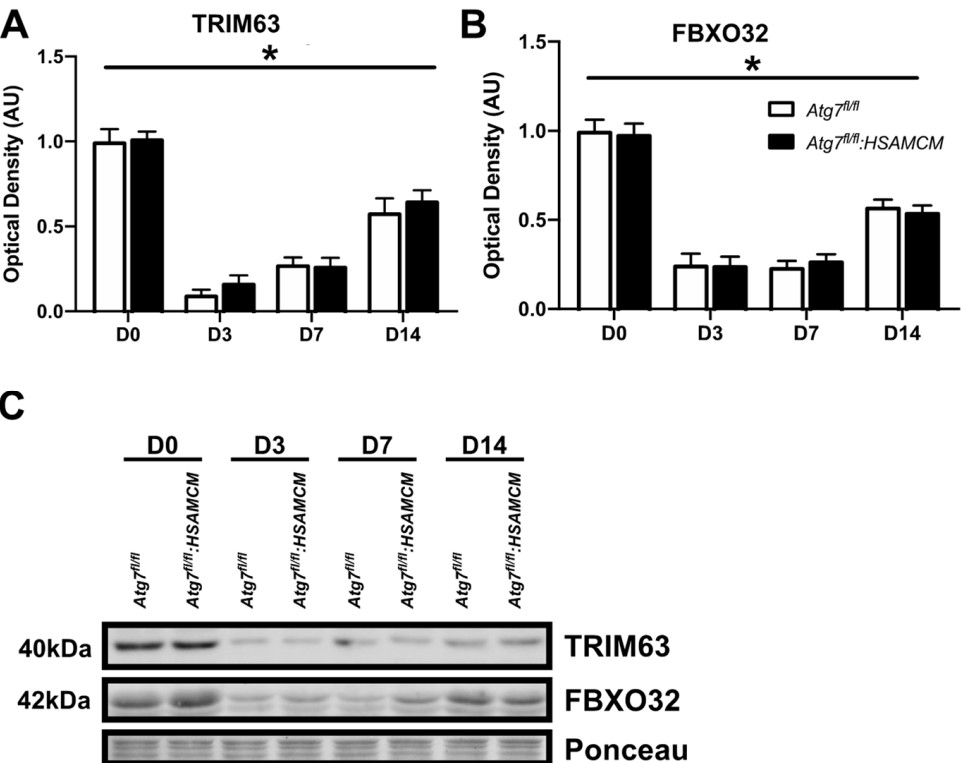

**Figure 6.** Atrophic markers in skeletal muscle before (D0) and following (D3, D7, D14) CTX-induced injury. (**A**,**B**) Quantification of TRIM63 and FBXO32 and (**C**) representative immunoblots. * Significant main effect of the regenerative time course ($p < 0.05$). n = 7–8 per group.

There were no significant apoptotic protein differences between *Atg7^{fl/fl}* and *Atg7^{fl/fl}:HSAMCM* mice. However, the pro-apoptotic protein BAX was elevated at D3 and D7 but returned to baseline levels by D14 (Figure 7A,H; main effect of time: $p < 0.05$). Interestingly, the anti-apoptotic BCL2 showed a similar pattern (Figure 7B,H; main effect of time: $p < 0.05$); however, no change in the BAX:BCL2 ratio was observed (Figure 7C,H). Similarly, the mitochondrial pore formation protein BID was several-fold higher in response to injury at D3 but decreased throughout the regeneration time course (Figure 7D,H; main effect of time: $p < 0.05$). In contrast, tBID was dramatically reduced at D3, but increased over the regeneration time course (Figure 7E,H; main effect of time: $p < 0.05$). Following injury, the tBID:BID ratio increased throughout the regeneration time course; however, it remained well below baseline levels (Figure 7F,H; main effect of time: $p < 0.05$). Finally, the anti-apoptotic protein XIAP was approximately sixfold higher at D3 compared to baseline and slowly declined until D14 (Figure 7G,H; main effect of time: $p < 0.05$).

*2.5. CTX-Induced Skeletal Muscle Damage Mediates a Transient Elevation in Apoptotic Enzyme Activity*

To further investigate degradative processes, we assessed several proteolytic enzymes. Although there were no significant differences between *Atg7^{fl/fl}* and *Atg7^{fl/fl}:HSAMCM* animals, there were transient elevations in CASP3, CASP8, and CASP9 at D7 before decreasing at D14 (Figure 8A–C; main effect of time: $p < 0.05$ for all caspases). Interestingly, CAPN and 20S proteasome activity were elevated at D7 but remained elevated until D14 (Figure 8D,F; main effect of time: $p < 0.05$). Of note, injured muscle displayed a 20-fold increase in cathepsin (CTS) activity at D3, which decreased over time (Figure 8E; main

effect of time: $p < 0.05$). Together, these data suggest a transient elevation in proteolytic enzyme activity in regenerating skeletal muscle.

### 2.6. Regenerating Skeletal Muscle Displays an Early Increase in ROS, Which Declines in Concert with Elevated Antioxidant Enzymes

Finally, we investigated the role of stress and antioxidant markers, along with total ROS production. HSP70, an important stress-related protein known to play an important role in the inflammatory response during muscle regeneration, was elevated in injured muscle compared to the uninjured muscle (Figure 9A,G; main effect of time: $p < 0.05$) but declined between D3 and D14. Antioxidant markers, including CAT, SOD1, and SOD2, were differentially expressed in regenerating muscles (Figure 9B–D,G; main effect of time: $p < 0.05$). CAT levels were elevated in injured compared to uninjured muscle, with peak levels at D7, after which it declined (Figure 9B,G). On the other hand, SOD1 and SOD2 levels were approximately 50% lower in injured compared to uninjured muscle (Figure 9C,D,G). However, SOD1 and SOD2 levels increased throughout regeneration. Furthermore, CYCS followed a similar pattern to SOD1 and SOD2 (Figure 9E,G; main effect of time: $p < 0.05$), hinting at an increase in mitochondrial content in the regenerating myofibers. In contrast, ROS generation (DCF) was approximately threefold higher in response to injury but declined over time (Figure 9F,G; main effect of time: $p < 0.05$).

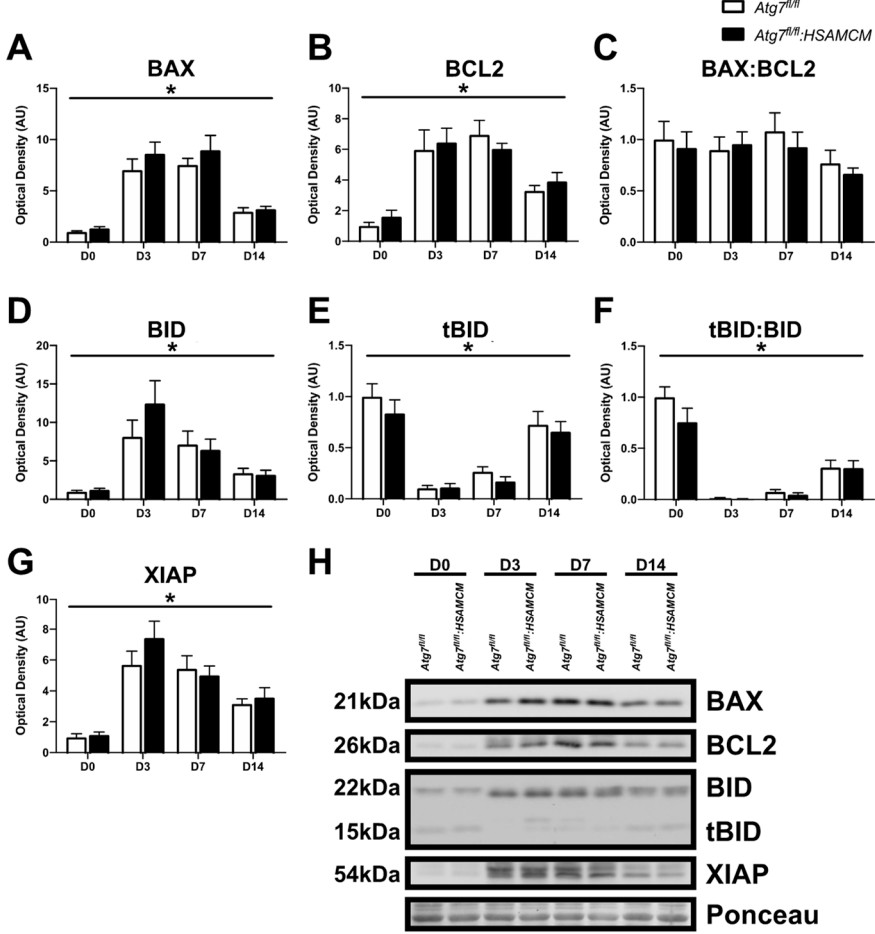

**Figure 7.** Apoptotic markers in skeletal muscle before (D0) and following (D3, D7, D14) CTX-induced injury. (**A**–**G**) Quantification of BAX, BCL2, BAX:BCL2 ratio, BID, tBID, tBID:BID ratio, XIAP, and (**H**) representative immunoblots. * Significant main effect of the regenerative time course ($p < 0.05$). n = 6–8 per group.

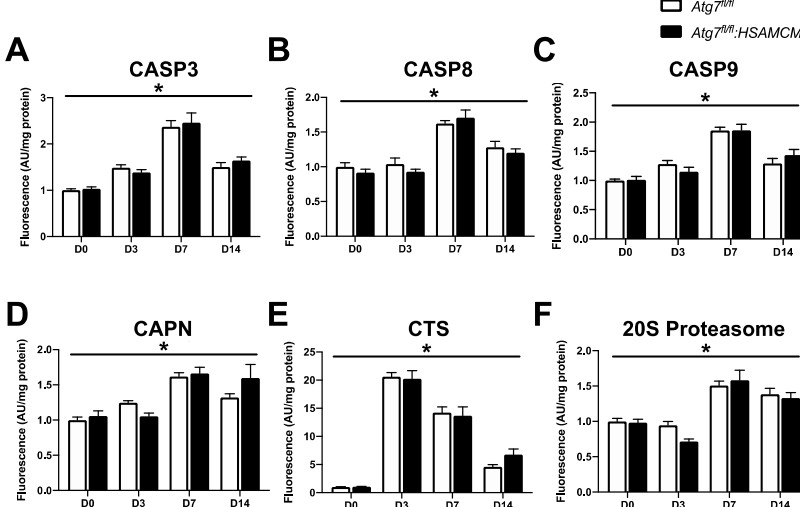

**Figure 8.** Proteolytic enzyme activity in skeletal muscle before (D0) and following (D3, D7, D14) CTX-induced injury. (**A–F**) Quantification of enzyme activity of CASP3, CASP8, CASP9, CAPN, CTS, and 20S proteasome. * Significant main effect of the regenerative time course ($p < 0.05$). n = 7–8 per group.

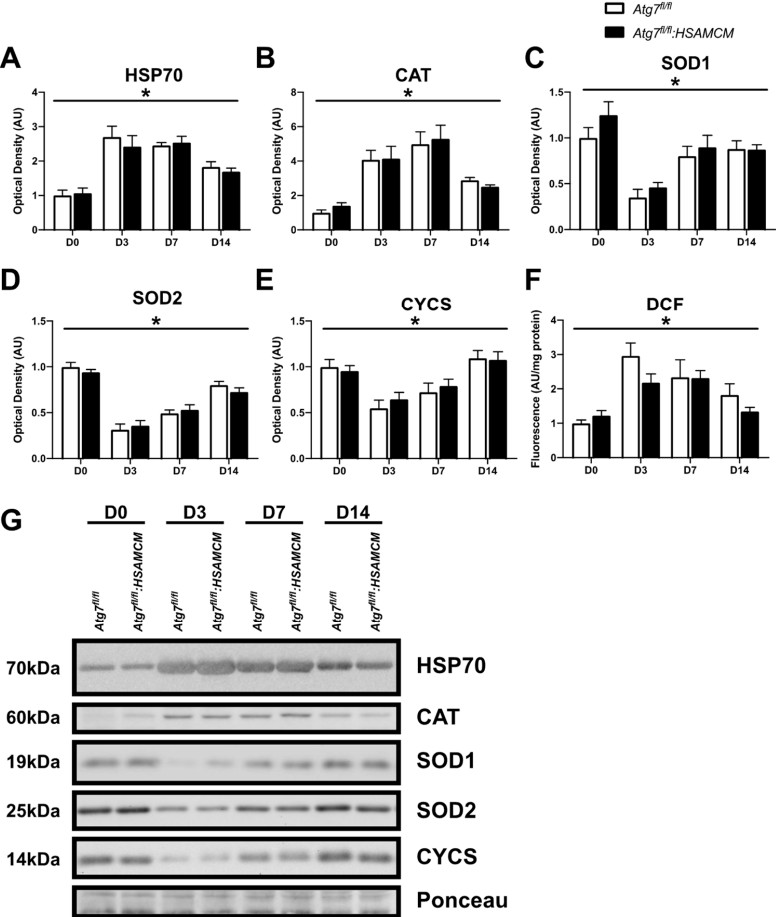

**Figure 9.** Expression of redox modifiers in skeletal muscle before (D0) and following (D3, D7, D14) CTX-induced injury. (**A–F**) Quantification of HSP70, CAT, SOD1, SOD2, and CYCS protein levels, as well as DCF fluorescence, and (**G**) representative immunoblots. * Significant main effect of the regenerative time course ($p < 0.05$). n = 6–8 per group.

### 3. Discussion

The goal of this study was to investigate the role of autophagy in regulating damage response mechanisms during skeletal muscle injury. Autophagy has been previously shown to mitigate apoptotic events and promote cell survival in skeletal muscle cells in culture [6,7] and muscle tissue in vivo [14]. It was originally hypothesized that autophagy deficiency in skeletal muscle would limit its regenerative capacity as a result of elevated apoptosis. Autophagy-deficient skeletal muscle displayed reduced muscle mass in an uninjured state, particularly in the gastrocnemius and plantaris. A reduction in autophagy-related proteins, including LC3II and the ratio of LC3II:I, was also observed in uninjured autophagy-deficient skeletal muscles. Skeletal muscle injury via CTX injection led to severe damage of muscle fibers and the restoration of ATG7 levels as early as D3. We suggest that the restoration of ATG7 levels is presumably attributed to the fusion of satellite cells containing a functional *Atg7* gene after CTX-induced injury. This is supported by data in D14 undamaged quadriceps muscles for knockout mice that showed a consistent depression in ATG7.

This study found a differential expression of degradative and apoptotic markers in regenerating muscle. In particular, we found that 3 days following CTX injection, skeletal muscle displays low levels of protein degradation markers TRIM63 and FBXO32. This is in contrast to a previous study that demonstrated greater levels of TRIM63 as early as D3 following CTX injury but lower levels of FBXO32 [16]. This difference may be attributed to the differences in volume and procedure of CTX injections, given this study [16] performed sequential injections equating to less than half the volume of CTX as the present study. Nonetheless, we found that TRIM63 and FBXO32 increased to approximately half of that found in injured muscles by D14 [16]. This may be relevant in the restoration of newly formed myofibers by limiting proteolysis and, thus, enabling myofiber growth [16]. The tBID:BID ratio, an indicator of apoptotic susceptibility [8], was slightly lower in uninjured autophagy-deficient muscles. Following injury, the tBID:BID ratio was significantly depressed and remained at approximately 40% of basal levels at D14. In contrast to these findings, the BAX:BCL2 ratio, a surrogate marker for mitochondrial outer membrane pore formation, remained largely unchanged throughout the regeneration time course. This is opposed to a previous report that demonstrated an elevated BAX:BCL2 ratio at D3 following CTX injury [17]; however, this previous study used approximately twice the dose of CTX for their injury model [17].

The proteins that form the core of the apoptotic machinery are the CASP. Data from the present study showed no changes in CASP3, CASP8, and CASP9 activity during autophagy deficiency in uninjured muscles. However, there was a time-dependent elevation in CASP3, CASP8, and CASP9, with a peak at D7 following injury. A previous report showed similar findings including greater total CASP3 and cleaved CASP8 at D3 [17]; however, this study injected twice as much CTX as the present study [17]. One of the major roles of CASP signaling is to degrade necrotic myofibers, as characterized in muscular dystrophy [18,19] and myofiber remodeling in atrophic conditions [20]. In addition, CASP3 plays a central role in the differentiation of myoblasts/satellite cells, a process that is essential for the regeneration of muscle [21,22]. Our lab and others have consistently shown peak CASP3 activity within 24 h upon the onset of differentiation in vitro [7,21–23]. However, the differentiation of satellite cells in vivo occurs after the inflammatory phase (~5 days following injury) [4,15,24–26]. Nonetheless, peak levels of CASP signaling at D7 may be attributed to the combination of the degradation of necrotic myofibers, remodeling of newly formed myofibers, and the activation/differentiation of satellite cells to aid in the regeneration of the muscle.

We also present data that showed no differences in CAPN, proteasome, and CTS activity in uninjured autophagy-deficient muscles. In addition, we demonstrate that CAPN and chymotrypsin-like activity of the 20S proteasome increase throughout the regeneration time course, particularly after D7. This is largely consistent with previous work showing peak *Capn1*, *Capn2*, and chymotrypsin-like activity at D3, which remained elevated up to

D14 following CTX-injury [27,28]. At rest, CAPNs are anchored to structural/scaffolding proteins such as titan in an inactivated form [29–33]. Disruption of the sarcomere scaffolding proteins via damage (e.g., exercise or injury) is thought to release and activate CAPNs to aid in the remodeling of the sarcomere by cleaving damaged myofibrils that undergo proteasomal degradation [29,31,34,35]. In addition, CAPNs indirectly function to repair the sarcolemma by cleaving dysferlin into mini-dysferlin C72, which forms lattices to stabilize the injured site and, thus, is important in myofiber membrane repair [36,37]. In contrast, CTS activity was significantly elevated at D3 and steadily declined through the regeneration time course. CTSs are a class of lysosomal enzymes that have previously been identified as important players in the injury response, the activation of which increases the degeneration of necrotic myofibers [17]. This is consistent with the decline in necrotic area within the muscle during regeneration. Taken together, there may be a shift from CTS/lysosomal degradation of necrotic myofibers that occurs early following injury, which is then followed by CASP/CAPN/UPS-mediated remodeling of satellite cells and regenerated myofibers.

Assessment of stress-response and antioxidant markers revealed an important transition that is linked to ROS generation. Three days following CTX injury, ROS generation in muscle homogenates was elevated in both genotypes. ROS generation and oxidative stress is a hallmark feature of the inflammatory phase of muscle repair [38] and is initiated by the infiltration of immune cells [38]. Neutrophils and pro-inflammatory macrophages are among the first to arrive in the damaged muscle [38,39]. They secrete several pro-inflammatory markers that not only increase ROS formation in neighboring immune cells but also within the injured muscle to accelerate the degradation of damaged myofibers [38–40]. Elevated ROS generation is also coupled with an elevation in HSP70 from baseline. Interestingly, HSP70 is an important protein that aids in regeneration by modulating the inflammatory response following injury and limits protein damage due to ROS [41]. In fact, the loss of HSP70 has been shown to be severely detrimental to muscle regeneration and results in a delayed and prolonged inflammatory response [41]. We detected greater levels of HSP70 at D3 post-CTX in both genotypes, which slowly declined throughout the regeneration time course. Concomitant with these findings, we found that ROS generation declines through the regeneration time course and is accompanied by elevated CAT, SOD1, and SOD2 levels. This coincides with the anti-inflammatory response that previous studies have identified [15,38,39], which is critical to neutralize excessive ROS and enable newly regenerated myofibers to grow into mature muscle [40]. Together, we provide data that link ROS generation and antioxidant enzymes to muscle regeneration.

An interesting finding from this study was the restoration of ATG7 within the injured skeletal muscle of our knockout animals. We demonstrated this restoration through immunoblots (Figure 2F,M) and immunofluorescence staining (Figure 3C) of ATG7 in injured skeletal muscle, despite continued knockdown in uninjured skeletal muscle (Figure 3A,B). We speculate that the restoration of ATG7 is likely attributed to the satellite cell population. This is probable as some residual ATG7 was found in skeletal muscle homogenate via immunoblot analyses (Figures 2F,M and 3A,B), with one source of this ATG7 likely being that from satellite cells. In agreement with this, no ATG7-positive staining was detected in D0 skeletal muscle fibers via immunofluorescence analysis (Figure 3C). These stem cells are central to skeletal muscle regeneration and contribute their nuclei to the existing muscle or fuse together to form new muscle [4]. It is likely that CTX injury "reset" the skeletal muscle because of the degradation of existing myofibers followed by its regeneration. As a result, the regenerated skeletal muscle fibers would contain the genetic material of the satellite cell [28,42–45] and, thus, functional *Atg7*, as knockdown would only occur in mature skeletal muscle fibers and not satellite cells due to the HSA promoter mouse model utilized [46]. Importantly, this contribution by the satellite cells may be a valuable approach to restoring or upregulating specific molecular mechanisms/pathways (i.e., autophagy) in pathophysiological states displaying alterations in these processes. These later data may support the use of satellite cell therapy to promote skeletal muscle regeneration through restored autophagic signaling in autophagy-compromised skeletal muscle.

## 4. Materials and Methods

### 4.1. Animals

Heterozygous C57BL/6 mice floxed for *Atg7* (*Atg7*$^{fl/fl}$; kindly provided by Dr. Herbert Virgin; Washington University, St. Louis, MO, USA) were bred with carriers of the skeletal-muscle-specific, tamoxifen-inducible mutated Cre under the control of the human alpha-skeletal actin promotor (HSAMCM; kindly donated by Karyn Esser; University of Kentucky Center of Muscle Biology, Lexington, KY, USA). Breeding of animals ultimately generated an inducible skeletal-muscle-specific *Atg7* knockout model (hereby *Atg7*$^{fl/fl}$:HSAMCM). *Atg7*$^{fl/fl}$ littermates were used as controls. To induce knockout, nine-to-twelve-week-old *Atg7*$^{fl/fl}$:HSAMCM and *Atg7*$^{fl/fl}$ animals were injected with tamoxifen for 5 consecutive days followed by a 2-week rest/washout period prior to induction of muscle injury. Following the 2-week rest period, animals were anesthetized with isoflurane and injected with 50 μL of cardiotoxin (10 μM; CTX; Naja mossambica mossambica; Sigma-Aldrich, Burlington, MA, USA; C9859) into the tibialis anterior (TA), and 75 μL to each head of the gastrocnemius of both legs (i.e., 6 injections total). Uninjured quadriceps muscles were also collected from animals. All mice received 60 μL subcutaneous injections of meloxicam following muscle injection and 24 h post-CTX. All animals had access to water and standard lab rodent chow ad libitum and were sacrificed by cervical dislocation, as approved by the University of Waterloo Animal Care Committee.

### 4.2. Skeletal Muscle Collection

Mice were sacrificed and tissues were collected 0 (before CTX injection; D0) as well as 3, 7, and 14 days following CTX injection (D3, D7, and D14). The TA was extracted from both legs: one was snap-frozen in liquid nitrogen for immunoblot and enzyme activity assays, while the other TA was embedded in optimal cutting temperature (OCT) compound (Tissue-Tek) and frozen in liquid-nitrogen-cooled isopentane, as previously demonstrated [14]. The gastrocnemius was also removed from each leg and was first separated into its two heads; then, each head was separated in half with equal portions of red and white gastrocnemius. The separated gastrocnemius along with the soleus, plantaris, and quadricep muscles were snap-frozen in liquid nitrogen. Muscle samples were stored at −80 °C until analyses.

### 4.3. Immunoblot Analyses

Gastrocnemius and quadriceps muscle were homogenized in ice-cold lysis buffer (20 mM HEPES, 10 mM NaCl, 1.5 mM MgCl, 1 mM DTT, 20% glycerol, and 0.1% Triton X-100; pH 7.4) that included protease inhibitors (Complete Cocktail; Roche Diagnostics, Mississauga, ON, Canada) using a handheld glass homogenizer [14]. Equal protein amounts from homogenates were loaded onto 12% SDS-PAGE gels and subjected to electrophoresis. Subsequently, the proteins were transferred onto PVDF membranes (BioRad, Mississauga, ON, Canada) and blocked either at 4 °C overnight or at room temperature for 1 h using 5% (*w/v*) milk-TBST. The blocked membranes were then exposed to the primary antibody at 4 °C overnight or at room temperature for 1 h. Following the primary antibody incubation, the membranes were washed with TBST and incubated with an appropriate horseradish-peroxidase-conjugated secondary antibody (Santa Cruz Biotechnology, Dallas, TX, USA) at room temperature for 1 h. The proteins were visualized using Clarity$^{TM}$ Western ECL Substrate (BioRad, Mississauga, ON, Canada) or ECL Western Blot Substrate (BioVision, Waltham, MA, USA) and a ChemiGenius 2 Bio-Imaging System (Syngene, Ottawa, ON, Canada). The optical densities of the detected bands were analyzed with GeneTools software (Syngene, Ottawa, ON, Canada). In order to ensure equal protein loading and transfer quality, a standard muscle sample with a known protein concentration was loaded into each SDS-PAGE gel for normalization purposes. Finally, to verify equal protein loading and the quality of protein transfer, the membranes were stained with Ponceau S (Sigma-Aldrich).

Immunoblots were incubated with primary antibodies against ATG7 (1:500; Cell Signaling, Whitby, ON, Canada; 8558), LC3B (1:500; Cell Signaling, Whitby, ON, Canada; 2775), BECN1 (1:1000; Cell Signaling, Whitby, ON, Canada; 3738), ATG4B (1:1000; Cell

Signaling, Whitby, ON, Canada; 5299), SQSTM1 (1:500; Progen, Wayne, PA, USA; GP62-N), HSP70 (1:2000; Enzo Life Sciences, Toronto, ON, Canada; ADI-SPA-810), XIAP (1:1000; Enzo Life Sciences, Toronto, ON, Canada; ALX-210-327), SOD1 (1:2000; Enzo Life Sciences, Toronto, ON, Canada; ADI-SOD-101), SOD2 (1:7500; Enzo Life Sciences, Toronto, ON, Canada; ADI-SOD-110), CAT (1:2000; EMD Millipore, Etobicoke, ON, Canada; 219010), CYCS (1:2000; Santa Cruz Biotechnology, Dallas, TX, USA; sc-13156), BCL2 (1:200; Santa Cruz Biotechnology, Dallas, TX, USA; sc-7382), BAX (1:1000; Santa Cruz Biotechnology, Dallas, TX, USA; sc-493), BID (1:500; Santa Cruz Biotechnology, Dallas, TX, USA; sc-11423), FBXO32 (MAFBX; 1:200; Santa Cruz Biotechnology, Dallas, TX, USA; sc-33782), and TRIM63 (MURF1; 1:200; Santa Cruz Biotechnology, Dallas, TX, USA; sc-32920).

### 4.4. Enzyme Activity Assays

The gastrocnemius muscle, utilized for caspase, calpain, cathepsin, and proteasome activity assessments, was homogenized in a comparable manner as previously described but without the addition of protease inhibitors. The resulting homogenates were then centrifuged at $1000\times g$ for 10 min at 4 °C, and the resulting supernatants were collected and stored at −80 °C. The protein concentration of these supernatants was determined using the BCA protein assay.

### 4.5. Caspase and Calpain Assay

Caspase and calpain activity assays were conducted following previously established protocols [7,47]. For caspase-3, -8, and -9 activity measurements in gastrocnemius muscle homogenates, specific substrates were employed: Ac-DEVD-AFC (Enzo Life Sciences, Toronto, ON, Canada; ALX-260-032), Ac-IETD-AMC (Sigma-Aldrich, Burlington, MA, USA; A4188), and Ac-LEHD-AMC (Enzo Life Sciences, Toronto, ON, Canada; ALX-260-080), respectively [14]. The muscle samples were incubated (in duplicate) in assay buffer (20 mM HEPES, 10% glycerol, 10 mM DTT) with the respective substrates at room temperature for 2 h. Subsequently, a Synergy H1 multi-mode microplate reader (Biotek, Mississauga, ON, Canada) was used to measure fluorescence with excitation and emission wavelengths set at 360 nm and 440 nm for AMC and 400 nm and 505 nm for AFC. Enzyme activity was then normalized to the total protein content, and the fluorescence intensity was expressed in arbitrary units per milligram of protein.

For calpain activity assessment, the gastrocnemius homogenate was incubated at 30 °C for 2 h in assay buffer (20 mM HEPES, 10% glycerol, 10 mM DTT) with the substrate Suc-LLVY-AMC (Enzo Life Sciences, Toronto, ON, Canada; BML-P802). In one additional well, the homogenate was incubated with both Suc-LLVY-AMC and the calpain inhibitor Z-LL-CHO (Enzo Life Sciences, Toronto, ON, Canada). Fluorescence measurements were taken using the Synergy H1 multi-mode microplate reader (Biotek, Mississauga, ON, Canada) with excitation and emission wavelengths of 360 nm and 440 nm, respectively. Calpain activity was determined by calculating the difference in fluorescence intensity between samples with and without the inhibitor and then normalizing it to the total protein content.

### 4.6. Cathepsin Assay

Activity of lysosomal enzymes (cathepsins L and B) was measured with the fluorogenic substrate z-FR-AFC (Enzo Life Sciences, ALX-260-129) as previously described [48]. The gastrocnemius homogenates were loaded into assay buffer (50 mM sodium acetate, 8 mM DTT, 4 mM EDTA, 1 mM Pefabloc; pH 5.0) in duplicates. Subsequently, the z-FR-AFC substrate was added to achieve an in-well concentration of 50 μM. Using a Synergy H1 multi-mode microplate reader (Biotek, Mississauga, ON, Canada) with excitation and emission wavelengths of 400 nm and 505 nm, respectively, the mixture was incubated at 30 °C for 30 min. Cathepsin activity was then normalized to the total protein content and expressed as fluorescence intensity in arbitrary units per milligram of protein.

*4.7. 20S Proteasomal Activity Assay*

The fluorogenic substrate Suc-LLVY-AMC (Enzo Life Sciences, BML-P802) was used to examine chymotrypsin-like activity of the 20S proteasome as previously described [49]. The gastrocnemius samples were incubated in proteasomal assay buffer (50 mM Tris/HCl, 25 mM KCl, 10 mM NaCl, 1 mM MgCl$_2$; pH 7.5) in the dark at 30 °C for 1 h, along with the substrate, or with a combination of the substrate and the proteasome inhibitor epoxomicin (Cayman Chemical, Ann Arbor, MI, USA; 10007806). Fluorescence was measured using a Synergy H1 multi-mode microplate reader (Biotek, Mississauga, ON, Canada) with excitation and emission wavelengths set at 360 nm and 460 nm, respectively. The 20S proteasome activity was determined by calculating the difference in fluorescence intensity between samples with and without the inhibitor and then normalizing it to the total protein content.

*4.8. Reactive Oxygen Species Generation*

Whole-muscle reactive oxygen species production was examined using dichlorofluorescein-diacetate (DCFH-DA; Life Technologies, Waltham, MA, USA; C6827) as previously described [50]. The gastrocnemius muscle samples were homogenized using ice-cold lysis buffer (250 mM sucrose, 20 mM HEPES, 10 mM KCl, 1 mM EDTA, 1 mM EGTA, 1 mM DTT; pH 7.4) containing protease inhibitors (Complete Cocktail; Roche Diagnostics, Mississauga, ON, Canada). The homogenization was carried out with a handheld glass homogenizer. For analysis, the samples were duplicated and loaded with 5 µM DCFH-DA. Afterward, they were incubated at 37 °C in the dark for 1 h. The fluorescence was quantified using a Synergy H1 multi-mode microplate reader (Biotek, Mississauga, ON, Canada) with excitation and emission wavelengths of 490 nm and 525 nm, respectively. The measured fluorescence intensity was then normalized to the total protein content and presented in arbitrary units per milligram of protein.

*4.9. Histological Analyses*

Ten-micron sections were cut from the TA for all histological and immunofluorescent analyses. Slides were first stained with hematoxylin for 30 s, washed, and counterstained with eosin for 3 min. Slides were washed and dehydrated in increasing concentrations of ethanol (70%, 95%, and 100%) before being cleared with xylene and mounted with a coverslip.

To quantify muscle damage, the area of the TA cross-sections that contained fibers with visible morphological damage or centralized nuclei was circled on days 3, 7, and 14. This regenerating area was then calculated as a percentage of the total injured cross-sectional area.

*4.10. Immunofluorescence*

Fiber-type staining was conducted in accordance with previously established protocols [14,51]. Briefly, ten-micron TA muscle sections were dried and blocked with 10% goat serum in 1× PBS for 1 h. A primary antibody cocktail against myosin heavy chains (MHCs) I (BA-F8; 1:25), MHCIIa (SC-71; 1:500), MHCIIb (BF-F3; 1:50), and dystrophin (MANDYS1[3B7]; 1:100) (Developmental Studies Hybridoma Bank, Iowa City, IA, USA) was then applied overnight. TA muscle sections were also used to stain for ATG7 (1:50; Cell Signaling #8558) and PAX7 (1:20; Developmental Studies Hybridoma Bank). In this case, sections were fixed for 10 min in 4% formaldehyde at room temperature, blocked in 10% goat serum in 1x PBS for 1 h, and incubated overnight with primary antibody. Subsequently, slides were washed and incubated with isotype-specific secondary antibodies (Life Technologies) for 1 h at room temperature. Slides were thoroughly washed, and coverslips were mounted using Prolong Gold antifade reagent (ThermoFisher, Waltham, MA, USA). Imaging was performed the following day using an Axio Observer Z1 fluorescent microscope equipped with an AxioCam HRm camera and associated AxioVision software (Carl Zeiss, North York, ON, Canada). Quantitative data for fiber-type composition was

determined by counting all fibers within a muscle section. Fiber-type cross-sectional area was determined by outlining fibers from 5 random regions of each section (i.e., ~60 fibers per muscle fiber type).

*4.11. Statistical Analyses*

Values were presented as mean ± SEM. Statistical analyses of uninjured muscle (D0) were performed using Student's *t*-test. A two-way ANOVA with a Tukey post hoc analysis was used to examine the effect of Atg7 knockout (i.e., genotype) and regeneration time point (i.e., day post-CTX). For all tests, $p < 0.05$ was considered statistically significant.

## 5. Conclusions

The findings from this study demonstrate that an initial skeletal-muscle-specific autophagy deficiency does not alter its regenerative potential, which, in this experimental design, was likely a result of the re-establishment of ATG7. We conclude that the contribution of satellite cells is likely responsible for the re-establishment of skeletal muscle ATG7 in our knockout mice, likely due to the addition of nuclei with functional *Atg7* gene. Consequently, similar damage, apoptotic, and degradative signaling responses were observed throughout regeneration in both genotypes. Overall, this suggests that the regenerative program that is largely contributed by satellite cells may adequately recover gene/protein expression and regenerative response in a rapid timeframe. Thus, satellite cell therapy may have the potential to not only promote skeletal muscle regeneration but also to restore autophagic signaling in autophagy-compromised skeletal muscle.

**Author Contributions:** Conceptualization, T.C. and J.Q.; format analysis, F.A.R., T.C. and D.B.; funding acquisition, J.Q.; investigation, F.A.R., T.C., D.B. and S.C.; methodology, F.A.R., T.C., S.C. and J.Q.; project administration, J.Q.; resources, J.Q.; supervision, J.Q.; validation, F.A.R., T.C., D.B., S.C. and J.Q.; writing—original draft, F.A.R. and T.C.; writing—review and editing, F.A.R., T.C., D.B., S.C. and J.Q. All authors have read and agreed to the published version of the manuscript.

**Funding:** This research was funded by the Natural Sciences and Engineering Research Council of Canada (grant number: 258590).

**Institutional Review Board Statement:** This study was conducted in accordance with the guidelines set by the University of Waterloo Animal Care Committee (AUPP#43315).

**Informed Consent Statement:** Not applicable.

**Data Availability Statement:** The data from the present study are available upon request from the corresponding author.

**Conflicts of Interest:** The authors declare no conflict of interest.

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
