# Peer review of "Degradative Signaling in ATG7-Deficient Skeletal Muscle Following Cardiotoxin Injury"

_muscles, doi:10.3390/muscles2030023_

Round 1

Reviewer 1 Report (Previous Reviewer 1)

The author's were mostly responsive to my previous feedback. New Figure 4B lacks error bars on the D14 data. These should be added prior to publication. Also, individual data points are still not displayed but should be added.

Author Response

Please see Attachment 

Reviewer 2 Report (Previous Reviewer 2)

Rhaman and colleagues have provided a revised and more compelling manuscript which addresses all of my concerns apart from one regarding the raw data around the western blots in Figure 1. Additionally, the authors are thanked for providing immunofluorescent labelling of ATG7 in muscle sections however, I have a new concern regarding the Pax7 labelling. Please see below for specifics.

1.       Regarding the western blots in Figure 1K, this reviewer appreciates that the authors may not want to include additional examples of the blots in the figure but as pointed out by my initial comments, in the interest of transparency, more replicates of the western blots should be provided. Please provide the raw western blots for review in a seperate file if necessary.

2.       Regarding the Pax7 IF label in Fig 3C, can the authors explain the diffuse expression of Pax7 in the HSAMCM mouse at D0? Pax7 positive satellite cells should provide more of a punctate signal ie from individual cells. This representative image shows very diffuse labelling which is not consistent with individual satellite cells. Are the authors certain this is not non-specific labelling?

3.        Line 259 - add a space between CSASP9 and activity

Author Response

Reviewer 3 Report (Previous Reviewer 3)

All revision points were satisfied no further comments. 

No major problem with the English. 

Round 2

Reviewer 2 Report (Previous Reviewer 2)

The authors have addressed my concern relating to the Pax7 IF labelling, thank you.

The authors are thanked for providing the raw western blots from Fig 1K however, upon reviewing these images, there appears to be an issue with how the data have been presented.

The labelling of samples/lanes for the LC3BI and LC3BII blots appears to have been mixed up. In the manuscript, LC3BII appears to be downregulated in the Atg7fl/fl:HSA-MCM mouse compared to the Atg7fl/fl. However, on the raw blot (gel 3) it appears that the alignment of the labels has been mixed up. The representative bands in the manuscript appear to be from the second and third lanes of gel 3 which corresponds to a D0 Atg7fl/fl:HSA-MCM sample and a D3 Atg7fl/fl sample respectively. This is also indicated by the artifact (a small dot) that appears above the LC3BI band in the Atg7fl/fl:HSA-MCM sample in the manuscript but on the raw blot, that artifact appears in lane 3, the D3 Atg7fl/fl sample. Can the authors please reconcile this? Given this error, the authors must provide all raw western blots associated with the manuscript and clearly label which bands have been used as representative examples of the data in the final manuscript.

Additionally, the densitometric analysis for LC3BII (Fig 1I) is significantly decreased compared to the control however, taking this mislabeling error into account, are those data still accurate? Looking at the raw blots provided, LC3BII does not really appear to change at the D0 timepoint, please revisit these data.

Finally, the authors indicate n= 7-8 samples per condition in the figure legend of Fig1. However, the raw blots show n=1 sample for the Atg7fl/fl and n=1 for the Atg7fl/fl:HSA-MCM per timepoint, across 4 gels. That equates to n=4 samples per condition. Please clarify the n numbers.

Round 3

Reviewer 2 Report (Previous Reviewer 2)

The authors have addressed all my concerns. 

This manuscript is a resubmission of an earlier submission. The following is a list of the peer review reports and author responses from that submission.

Round 1

Reviewer 1 Report

This is an original manuscript that seeks to understand the relationship between apoptosis and autophagy in the context of skeletal muscle regeneration. The paper is well-written, uses a unique muscle-specific knockout model, and the techniques and biochemical quality of the data meet or exceed current standards in the field. However, it is not clear how applicable the conditional knockout model is to muscle regeneration and some of the conclusions extend beyond the results. Specific comments are provided below.

-          Some background on Atg7 needs to be included in the introduction. The results and mouse model are difficult to interpret without some background as to what Atg7 does in relation to autophagy. Similar comments would apply to other autophagy markers used in the Results.

-          Please show the individual data points on the graphs.

-          Error bars are lacking for Figure 3B, D14.

-          The Atg7 knockout appears to be specific for skeletal muscle fibres but not MuSCs. Since MuSCs are responsible for skeletal muscle regeneration, the applicability of this model to regeneration needs to be addressed. Why was the Atg7fl/fl not crossed with Pax7cre?

-          Font size and line spacing are inconsistent in the Discussion (see lines: 246-249 as an example).

-          Line 291-293: This conclusion stretches beyond the data. The article provides evidence that markers of oxidative stress coincide with the muscle regeneration time course; however, sufficient evidence to label them ‘molecular markers’ of regeneration is lacking.

-          Lines 301-305: This conclusion is a stretch beyond the findings. Restoring gene expression following myofibre-specific ablation in regenerated muscle is an artifact of the model. In clinical situations, cell-specific knockouts or defects that would only affect myofibres and not MuSCs would be rare. Further, gene therapy via MuSCs has been attempted in other conditions (i.e., models of muscular dystrophy) and there is insufficient MuSC engraftment for meaningful restoration of gene expression.

-          Why was meloxicam used following CTX injection? Are there any off-target effects in skeletal muscle?

- Quadriceps dissection is not described but used in Western Blotting (See Methods).

Reviewer 2 Report

Rahman and colleagues describe the use of well-established protocols and resources to demonstrate that muscle fibre specific knockout of Atg7 does not affect skeletal muscle regeneration or apoptosis in response to cardiotoxin injury in mice. Overall, the science is well executed however, clarification around certain experimental controls and methodologies is required. Most importantly, this reviewer feels that providing greater insight into the source of the Atg7 upregulation in response to CTX administration in the HSAMCM is needed to support the conclusions that Atg7 is being derived from non-muscle cell populations or from satellite cells. Please see below for further details.

Major comments

1.       The overall narrative of this study describes the loss of Atg7 leading to changes in apoptosis in the context of skeletal muscle regeneration. The authors demonstrate that ~40% of total Atg7 remains in muscle homogenates in the flox HSAMCM mice. While the authors suggest that the source of this Atg7 is other cell types residing in the muscle, it is critical this is actually shown to be the case. Can the authors perform immunofluorescence studies to show that no Atg7 is present in myofibres of HSAMCM flox mice or show that the Atg7 that is present is not in myofibres? Alternatively, the authors could isolate primary fibres from these mice and western blot a pure population of myofibres to address this concern.

2.       Have the authors accounted for any effects of Cre on muscle regeneration. The control used here is the Atg7fl/fl but this mouse does not express Cre. Is there existing literature that can be cited to demonstrate that Cre mice do not display an altered regenerative phenotype compared to wildtype mice?

3.      Figure 1K – the authors show only n=1 per group for western blot analysis. In the interest of transparency and given 7-8 mice were used per group, please include more biological replicates on the western blot.

4.       The authors show western blot data from GAST muscles in Figure 1, QUAD muscles in Figure 2 and histology from TA muscles in Figure 3. Why did the authors not use all muscle groups in these figures or only analyse one muscle group consistently? Are the findings consistent across muscle groups?

5.       Can the authors demonstrate that Atg7 is or is not expressed in the satellite cells of the Atg7HSAMCM mice that received CTX? Does the HSAMCM promoter also act in satellite cells? Is there a reference that can be provided to address this? Alternatively, isolation of satellite cells from injured and uninjured Atg7fl/fl and HSAMCM mice followed by western blot for Atg7 is necessary to address this point.  

6.       Methods – the authors state that 30 muscle fibres per fibre type were analyse for CSA. This accounts for ~5% of the total number of muscle fibres in a TA muscle. This reviewer believes this is under-sampled. At least 10% of the total number of fibres should be measured for CSA analysis. The authors are potentially missing information by not measuring the size of more fibres. Please address. Additionally, the authors state that whole muscle CSA was measured, why was this and to what data does it relate to? Whole muscle area is very easily affected by a number of technical variables such as consistency in dissection, quality of cryosectioning etc. Please clarify.

Minor comments

1.       Line 8 – remove comma after ‘given’

2.       From reading the text, it is not clear if the mice in Figure 1 have been treated with tamoxifen or not. Please clarify.

3.       For clarity, it would be better to describe day 0 mice as uninjured throughout the entire manuscript.

4.       The authors do a good job of explaining the role of the analysed targets in autophagy or apoptosis in the discussion. However, previous to the discussion section, the rational for why these targets were analysed is not clear. For a reader outside the autophagy space, this could be difficult to follow conceptually. Perhaps consider including the roles and rational for analysing these markers earlier in the manuscript.

5.       Figure 2M, 5C and 8G – the D0, D3, D7 and D14 markers do not align up with the appropriate lanes of the western blot, please correct.

6.       Can the authors please clarify in the methods section whether or not the D14 uninjured Atg7HSAMCM mice received a sham injection?

7.       Line 246 – please correct other to others

8.       There appears to be a formatting issue with the size of the citation call outs in the Discussion section.

9.       Methods – the authors describe the loading of a standard muscle sample of known protein concentration for normalisation however, this does not account for variability in loading on the gel itself. Have the authors normalised protein levels to total protein (Ponceau) to account for variability in loading? Please clarify in the methods section.

Reviewer 3 Report

In the study described in the manuscript, Authors Rahman et al., generated a ATG7 skeletal muscle specific knock-out line driven by skeletal actin promotor (ATG7fl/fl:HSAMCM) in order to study the function of ATG7 in muscle regeneration. When they were carrying out the studies, they initially observed phenotype of ATG knockout in muscle as body weight and muscle weight of GAS and PLANT were less compared to the wild type mice. At the same time, LC3B-II:I ratio decreased indicating the autophagic flux was impaired. However, when they further conducted the muscle injuries, the autophagy was restored and no abnormal muscle phenotype was recorded. The authors hypothesized that the restore the autophagy was caused by satellite cells which express ATG7 participated in the muscle regeneration and restored the ATG7 expression. The authors switched the direction of the study to examine the oxidative stress and apoptosis change during regeneration time course which deviate from the original study goal. Although, understanding the above elements in muscle regeneration is important, the study has very little relevance to ATG7. The design of the study was flawed as tamoxifen was only given at the very beginning of the study but it should have been given right before the muscle injury too. Therefore, the study does not carry enough merit unless the authors repeat the in vivo study with administration of tamoxifen before injury or during the course of regeneration.

Minor issues:

1.       The title is not appropriate and does not describe the study.

2.       In figure 1, there was no description of which muscle was used to generate the western blot.